# How Much Money Do You Need to Feel Taller? Impact of Money on Perception of Body Height

**DOI:** 10.3390/ijerph18094533

**Published:** 2021-04-24

**Authors:** Radosław Walczak, Przemysław Zdybek, Felice Giuliani, Luca Tommasi

**Affiliations:** 1Psychology Institute, University of Opole, 45-040 Opole, Poland; pzdybek@uni.opole.pl; 2Department of Psychological, Health & Territorial Sciences, University of Chieti-Pescara, 66100 Chieti, Italy; felice.giuliani@unich.it (F.G.); luca.tommasi@unich.it (L.T.); 3Department of Neurosciences, Imaging and Clinical Sciences, University of Chieti-Pescara, 66100 Chieti, Italy

**Keywords:** body height, money impact, height estimation

## Abstract

Body height is considered to be one of the most important reproductive signals. However, there are only a few publications on what influences the sense of whether we assess ourselves as tall or short. In the present contribution, the psychological impact of money on the evaluation of a person’s own height was tested. We performed two experimental studies in which the respondents had contact with different amounts of money and were asked to evaluate their body height with the use of a laser pointer. The first experiment (*N* = 61) showed that contact with money significantly increased subjective height evaluation, and the effect was independent of participants’ real body height. The second experiment (*N* = 120) replicated the effect of money manipulation. Moreover, it was shown that higher amounts of money increased one’s own height estimation more than smaller amounts. Our research shows that money can be used for building one’s social position, which is an attractiveness signal that can influence one’s own height evaluation.

## 1. Introduction

A person’s height appears to be one of the key attributes in determining interpersonal attractiveness [1,2,3]. There are significant differences in height between genders [4] that affect mate selection preferences. Men are taller and women prefer taller men as romantic partners [2,5]. Despite this difference, a preference for a slightly taller silhouette can be observed for both genders across many cultures [6]. Besides evolutionary principles, there are also social arguments for greater posture preferences. In many languages, being tall is equivalent to having power, as indicated in the English phrases ‘high position’ and ‘tower above others’, which indicate both social power and high posture. It is worth noting that a positive correlation between estimated height and social status was found in some studies [7,8,9,10]. These results suggest that there is a cognitive mechanism that connects observed social power with a distortion of people’s perceptual judgements of height. This constructivist theory implies that individuals actively build the perception of the world based on their motivations, expectations, and prior experiences [11,12,13].

There are also some examples of misrepresentation between real and self-presented height observed in the dating context, both in printed personal adverts [14] and in online dating profiles [15]. The same phenomenon is probably also related to the tendency of women to optically increase their leg length by wearing high heels [16], which also influences their self-assessed attractiveness levels [17]. Importantly, a preference for longer legs can be observed in both genders [3]. Taken together, it can be assumed that there is a tendency to look taller that is present in both genders. A taller appearance is related to a greater perceived social status, which is also interpersonally attractive, especially for women [18]. But the tendency to actively construct one’s biased height perception might require a specific contextual trigger. One such trigger will be discussed in the next paragraph.

### 1.1. Money as a Self-Oriented Social Distinctiveness Trigger

Money is the basis for the exchange of goods between people. Over time, it gained an additional symbolic meaning. Nowadays money is an indicator of status [19,20], but also an extension of the self [21,22]. Recent studies show how it can change our behavior and self-attitudes [23].

Experiments carried out by Vohs and colleagues [24] showed that activating thoughts about money increases the focus on one’s own goals but also decreases the will to share resources with others. Another study [25] reported that an increase in thinking about financial motives was positively related to seeing oneself as special and increased the need to show off. These effects may be important from an evolutionary perspective. Both men [19,26] and women [27] tend to use money and its derivatives to improve their social image. Some authors even suggest that some high-value consumer goods, which can be purchased with a significant amount of money, can be interpreted as costly signals that genuinely increase the mating value of the presenter [20]. This agrees with the results of Griskevicius et al. [28], showing that spending money was related to inducing mating motives.

Additionally, an overview of money research [23] suggests that money, in general, is an embodiment of social distinction, which means that money works as physical proof of social status, with all its consequences. Specifically, it is suggested [23] that any money activation elevates the person’s perceived social status relative to others, which should translate to an increase of all kinds of personal self-evaluations, especially those related to social position, including the person’s height and also interpersonal attractiveness, which is directly related to height.

What can be concluded from the above-mentioned studies is that money may be related to interpersonal attractiveness and therefore may trigger a state where one’s own height perception can be biased towards a taller, more attractive one. The effect should work especially well for higher amounts of money, as this allows more luxurious goods to be purchased [29], and higher nominal value generally increases perceptual distortions more [30].

### 1.2. Effectiveness of Different Money Denominations

It is difficult to judge how people evaluate the value of money of different denominations. The money illusion effect suggests that people misjudge the real value of money and rely more on its nominal value [31]. The higher the nominal value, the more positive the perception of the value of money (regardless of the underlying real value; but see [30,32]). This would mean that people would value the money of higher nominal value more than money of similar real value (in different currency) but with lower nominal value. This effect was measured with prices in Euro and Swedish crowns (one Euro equaling approximately nine Swedish crowns) [33]; the lower nominal Euro was perceived as disproportionally weaker than its real value would indicate. A recent study [34] shows that the value of the banknote influences both the emotion and cognition of the evaluating person, with stronger effects from a greater nominal value.

The above-mentioned studies describe a relation between the nominal value of money and its evaluation: the higher the nominal value, the greater its perceived worth. Such an effect could be anticipated in the present study as well.

### 1.3. Summary and Hypotheses

The reviewed literature suggests a relation between feelings of power and social status, money, height, and interpersonal attractiveness. Specifically, a higher-than-average silhouette is perceived as more attractive by males and by females [6]. Similarly, a higher social status is perceived as more attractive than lower status [18]. On top of that, money may be treated as a signal of social status [23] and therefore, also as a boost to attractiveness evaluations. Additionally, money biases self-evaluation [26], increasing the tendency to overestimate one’s value. Taken together, these ideas allowed us to speculate that money will trigger a need to boost one’s attractiveness in the form of one’s own height representation. There is also evidence that this effect should be proportional to the nominal value of money [30,34]. These considerations can be phrased in the form of the following three hypotheses: (H1) money increases one’s own height evaluation; (H2) money of higher nominal value increases one’s own height evaluation more than money of lower nominal value or no money at all; and (H3) money of higher nominal value increases evaluation of one’s own height more than money of comparable real value, but with lower nominal value.

## 2. Study 1

The principal aim of the first study was to test the main hypothesis, namely that money acts as a trigger that boosts one’s own height self-assessment.

### 2.1. Participants and Procedure

A total of 61 volunteer undergraduates took part in the study (20 males and 41 females) and they received no monetary compensation. The participants’ mean age was 21.23 years (SD = 3.83). All the experimental procedures were conducted individually, with each participant accompanied by a research assistant who was blind to the research procedure.

Each participant was informed that one of the purposes of the study would be to measure their mathematical abilities and hand–motor coordination. Thereafter, she or he gave written, informed consent for taking part in the research. After signing the forms, the participant was randomly assigned to one of two groups. Participants in both groups read the same cover story about mathematical abilities and hand–motor coordination measurement. Subsequently, the procedure differed according to the experimental condition.

In the experimental condition (*N* = 30), the participant was handed a small pile of money consisting of five banknotes with a nominal value of 20 PLN each (100 PLN in total). She or he was asked to count the notes and state the total amount. In the control condition (*N* = 31), participants were not handed money and did not have to count anything. Thereafter, the procedure was identical for both the experimental and control groups. Each participant was asked about their handedness and was given a laser pointer to the dominant hand. Then, standing 3 m from a blackboard (which was 27 cm wide and 210 cm high), the person was asked to indicate his or her height on the board using the laser pointer. All participants were instructed to hold the laser pointer at the level of their hip, with the arm outstretched along the body. Such a procedure was used so that the person could not use their body or arm as a reference point to estimate their own height. Once the pointer stabilized, the research assistant marked it on the board and measured the distance from the floor to the marked point. Afterwards, all respondents filled out a short questionnaire with their sociodemographic data, and their real heights were measured by the research assistant with the same measurement device as used for the marked height estimate. Finally, each person was sent to an adjacent room where they were informed about the real purpose of the study. It was accompanied by an explanation of the reasons for not giving this information at the start of the study. No negative reactions from the participants were observed at this point. In the end, each person was thanked for their participation.

### 2.2. Results

The participants’ mean height measured by the research assistant at the end of the experiment was 171.72 cm (SD = 9.77), whereas the self-assessed height was 173.83 cm (SD = 10.66). We found no difference in real height, between research and experimental groups (independent height measurement by experimenter’s assistant, *F*(1,59) = 1.03, *p* = 0.315, partial *η***^2^** = 0.017).To verify the primary hypothesis, we first calculated the difference between the person’s real height and that estimated with the laser pointer. Then we checked the distribution of the height estimate’s difference separately for the experimental and control groups with the help of Shapiro-Wilk’s W test. As the dependent variable distribution proved to be no different from the normal distribution (Shapiro-Wilk’s *W* = 0.965, *p* = 0.395 for the control group; *W* = 0.970, *p* = 0.545 for the experimental group), an analysis of variance (ANOVA) model was used to check for the main effect of the money activation while controlling for the gender effect. There was a main effect of money on the estimated height (*F*(1,57) = 9.394, *p* = 0.003, *η***^2^** = 0.141), which proves the main hypothesis: participants in the money group evaluated themselves as taller than those in the control group. The effect of gender was not significant (*F*(1,57) = 0.99050, *p* = 0.324, *η*^2^ = 0.017), which proves that both males and females overestimated their height by a similar amount. However, contrary to our assumptions, we found a significant interaction effect (*F*(1,57) = 5.526, *p* = 0.022, *η*^2^ = 0.088), which indicates that gender modified the impact of money on one’s own height estimation. Specifically, for men, money activation worked as assumed, but for women, there was no height estimation difference between the experimental and control groups. The details for the interaction are shown in Figure 1 and Table 1.

To verify if the money actually increased the height estimation both for men and women, we checked if the difference between real height and estimated height was greater than zero in each group through a single-sample *t*-test. The results are presented in Table 2.

As can be seen from Table 2, there was a significant difference (overestimation) both for males and females from the money activation (experimental) group and no difference from zero (no overestimation) for either males or females in the control group, which finally proved H1.

## 3. Study 2

The second study aimed to verify if money has a positive impact on one’s own height evaluation (H1), if the effect size increases with the stimulus size, namely the amount of money (H2), and if money of higher nominal value increases the evaluation of one’s own height more than money of lower nominal, but higher real, value (H3).

### 3.1. Participants and Procedure

A total of 120 undergraduate volunteers from extramural studies took part in this second study (42 males and 78 females). Participants’ mean age was 40.32 years (SD = 11.94). Participants were recruited at the university campus and randomly assigned to one of four research groups. Respondents from all groups signed an informed consent form to take part in the research. In the beginning, all participants read the same cover story as in Study 1. Afterwards, the procedure differed according to the experimental condition. In the three experimental groups, the first task for the participants was to count a given amount of money. The control group did not receive any money to count. Variants of cash to count were as follows: 100 Euro, consisting of five banknotes with a nominal value of 20 Euro each (100 Euro equaled about 430 PLN at the time of this research, an exchange rate generally known by most educated Polish respondents); 400 PLN, made up of ten banknotes with 20 PLN nominal value and four banknotes with 50 PLN nominal value; and 200 PLN, comprising five banknotes with 20 PLN nominal value and one banknote with 100 PLN nominal value. After counting the money (or, in the case of the control group, directly after reading the cover story), the research participants in all groups indicated their height on a board with the use of a laser pointer, as in Study 1. The remaining procedure was also identical to that in Study 1. 

### 3.2. Results

The mean real height measured by the research assistant at the end of the research procedure was 170.44 cm (SD = 9.50 cm) whereas the mean estimated height was 171.47 cm (SD = 11.53 cm). We found no difference in real height between groups, similar to Study 1 (*F*(3,116) = 0.78, *p* = 0.506, *η***^2^** = 0.019).

To confirm the main hypothesis and verify the other assumed hypotheses, an ANOVA test approach with the research group as an independent variable (control; 200 PLN nominal value; 400 PLN nominal value; 100 Euro nominal value) was used, while also assessing the effect of gender. The dependent variable was the height bias in centimeters (difference between real measured height and height indicated by the laser pointer). The analysis shows that there was no gender effect (*F*(1,112) = 0.003, *p* = 0.954, *η*^2^ < 0.001). The main effect of money manipulation was significant (*F*(3,112) = 14.520, *p* < 0.001, *η*^2^ = 0.280). There were also no significant interaction effects (*F*(3,112) = 1.458, *p* = 0.230, *η*^2^ = 0.038), which means that in this study money manipulation worked similarly on males and females. The between-group comparisons and effect sizes are presented in Table 3.

The results shown in Table 3 provided the second confirmation of H1—money activation increases one’s own height perception. There was, however, one caveat—the effect did not work for a foreign currency of low nominal value.

The data in Table 3 also proved H2: that money with higher nominal value increases one’s own height evaluation more than money of lower nominal value or no money at all. The effect size for a greater amount of money (400 PLN) was larger than that for a smaller amount (200 PLN).

Lastly, the data did not confirm H3: that money with higher nominal value increases the evaluation of one’s own height more than money of comparable real value but lower nominal value. On the one hand, the money effect was observed for both the 200 PLN and 400 PLN nominal values but not the 100 Euro (with a higher real value of approximately 430 PLN), which partially supported the hypothesis. The higher nominal value was related to a stronger positive height estimation bias. However, the lack of any money effect from the 100 Euro nominal value did not allow for full hypothesis confirmation. The money of lower nominal value but higher real value did not cause any change in height evaluation.

## 4. General Discussion

To our knowledge, there has been no other research that shows how money activation influences the perception of one’s own height. We were able to show that money activation could trigger a social distinctiveness motive, which would explain why people would want to perceive themselves as taller. According to the presented research, the effect of money changes the estimation of one’s own body height—one of the attractiveness signals important for both genders. Possessing or, as in our study, just holding money induced a state where people evaluated themselves as taller. If this is, as we assume, related to the attractiveness boost, it is possible that it could work on other attractiveness dimensions: for example, strength in males or youth in females. This would require further experiments to confirm.

Another explanation that we considered treats money as a social signal that directly influences status perception [23], which in turn is also related to height perception bias [7,8,10]. As social status and attractiveness are related [35], both lines of reasoning support each other and are consistent with the results observed in the study.

The presented money effect showed that larger amounts of money (400 PLN) have a stronger effect than smaller amounts (200 PLN). There were not enough different nominal values to estimate whether height increases linearly with nominal value or, as suggested by Manippa et al. [32], increases logarithmically. Nonetheless, an additional comparison between studies in the current research corroborated the hypothesis of Manippa et al., as the effect for 100 Euro was even smaller than that for 200 PLN. This further suggested that nominal value plays a crucial role in the effect, as the real value of 100 Euro should be approximately double, whereas its effect was in fact smaller. The effect sizes did not double with the doubling of nominal value, which would suggest a rather logarithmic relation. It needs to be mentioned that the nominal values of different amounts in various experimental conditions were not achieved with the same banknotes (for example the amount of 200 was composed of two times the 100 note, and 400, of four times the 100 note). We used a more ecologically true mixture of banknotes, which could have had some impact on the results.

A slight complication for a comparison between our two studies was the age or mode of study that differentiated the respondents in our samples. In Study 1, we tested young undergraduates who did not have to pay for their studies and only sometimes worked part-time. In Study 2, we tested students taking extramural courses who needed to pay for their studies and usually worked full-time (their classes took place on weekends). Thus, the amount of money that worked in Study 1 (100) might not have been enough to trigger an effect for the participants in Study 2; therefore, we cannot be sure at which level the money activation effect works. We know that the method has some limitations because a nominal value of 100 in a foreign currency did not work in Study 2, which we believe may have been related to a lower emotional value attached to the foreign currency (in our study, Euro) compared with the local currency (PLN). This line of reasoning was present in other studies [32,34], where it was the level of emotional attachment that differentiated the effectiveness of various stimuli. However, this explanation would also require further studies where the level of emotional arousal would be controlled.

There were some other limitations to our study. The sample consisted of university students, a group that is usually not on top of the social ladder. We believe that a more diverse sample (especially people with higher social status or more personal income) could help to clarify the limits of the effect obtained. Additionally, giving money or earning it (versus only holding it) could be more effective. Wang et al. [23] indicated that the more vivid the money activator, the stronger the effect, but due to the limited budget in our study, we could not afford directed gratification for the research participants. This distinction could also be clarified in future studies. A further limitation of this study could have been the spatial abilities level of our participants, which was not controlled. Future replication should include controlling this trait as well.

## 5. Conclusions

This paper presented two studies concerning the influence of money activation on the evaluation of one’s own height. We demonstrated that in the experimental condition, the participants changed their own height perception following contact with physical banknotes. This was possibly related to a social distinctiveness motive, which was the main theoretical mechanism assumed for the process. However, we must state that the proposed mechanism was just one of several possible candidates. Some alternative mechanisms would include the treatment of money as a social signal directly boosting height estimation, or other mechanisms we did not consider. The main research finding concerned the impact of money activation on one’s own height perception. This effect seems to work for both genders, although in one of our studies it was stronger for males. The second study, besides replicating the main finding, showed that money with higher nominal value increased one’s own height evaluation more than money of lower nominal value. In other words, contact with larger amounts of money had a larger impact on the overestimation of one’s own height. The observed effect worked only in the case of money in the local currency.

The achieved results corroborated the idea that money is cognitively processed as a symbol of success, social distinction, and probably attractiveness. Therefore, we can say that these abstract representations have been internalized, becoming part of our body representation [21]. Such an occurrence suggests that money may have a bidirectional relationship with the body, which is in line with the suggestions from Oullier and Basso [36]. It can be speculated that this is the result of pervasive social learning that repeatedly exposes us to the idea that money is related to success [37]. In our society, money is often a reward for being good at something, and since it is a scarce resource, a person must prove some kind of ability to make money (but see also [19]). Nevertheless, money is a signal that some primary achievement has occurred in the first place. 

However, money can also become completely disengaged from primary achievements and be directly processed as a symbol of success, as may have been the case in our experiments, in which participants did not receive money as a result of a specific action or choice. 

This implies that what counts is holding money, no matter where it comes from. Taking that implication a step further, this type of motivation toward money may sometimes foster dishonesty, greed, and corruption in politics and business [38]. In fact, people often tend to find shortcuts to make money easily and effortlessly, sometimes legally as in the example of buying lottery tickets, otherwise not. In particular, dishonest ways of obtaining money may be detrimental to our society, environment, and even our economic system. To provide some examples, we can think of financial speculation and related economic crises, or how corruption negatively affects politics and society. 

Finally, it needs to be stated that the results of our study are preliminary and require further research for final confirmation. Some especially interesting directions for future research, next to those mentioned above, would include determining the shape of the relation (linear or logarithmic); application of the effect to different forms of money, other than cash (e.g., virtual money, e-currency, earnings); and the impact of money on different dimensions of attractiveness and self-evaluation (financial perspectives, physical attractiveness, interpersonal attractiveness, self-esteem, intelligence, etc.), and the entire money-body bidirectional relationship.

## Figures and Tables

**Figure 1 ijerph-18-04533-f001:**
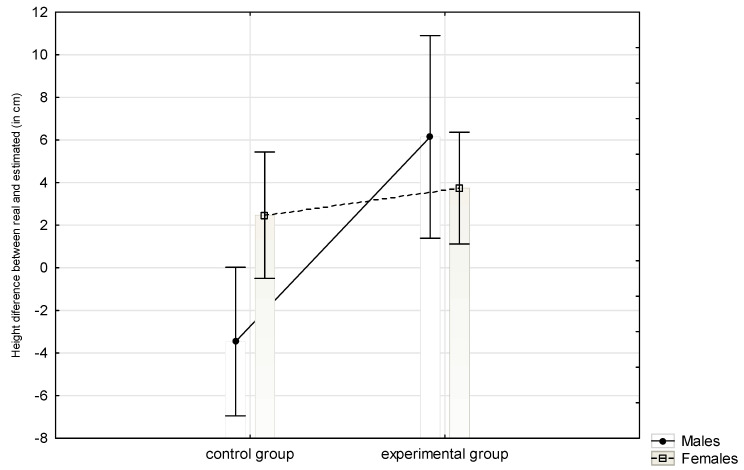
Interaction effect between experimental condition and gender. The dependent variable is the height difference (perceived body height minus real height measured by a research assistant).

**Table 1 ijerph-18-04533-t001:** The height difference between control and experimental groups.

Research	*N* (Male; Female)	Height Difference		Confidence Interval		Effect Size
Group		M	SD	−95%	95%	(Cohen’s *d*)
Control	31 (13; 18)	0.068	4.352	0.781	−1.528	
Experimental	30 (7; 23)	2.539	3.267	0.596	1.320	0.642

**Table 2 ijerph-18-04533-t002:** The height overestimation difference from zero.

Gender	Group	M	SD	*n*	Standard Error	*t*	df	*p*	Effect Size (Cohen’s *d*)
Male	Control	−3.46	7.31	13	2.03	−1.70	12	0.114	
Female		2.47	6.87	18	1.62	1.52	17	0.145	
Male	Exp.	6.14	3.13	7	1.18	5.18	6	0.002	1.706
Female		3.73	5.81	23	1.21	3.08	22	0.005	0.199

**Table 3 ijerph-18-04533-t003:** Mean height differences and comparison to the control group.

Research	N	Height Difference			*t*-Test		Effect Size
Group	(Male; Female)	M	SD	*t*	df	*p*	(Cohen’s *d*)
Control	30 (9; 21)	−2.267	7.390				
200 PLN	30 (11; 19)	2.933	2.180	3.696	58	0.000	0.954
400 PLN	30 (11; 19)	5.933	5.747	4.797	58	0.000	1.239
100 Euro	30 (11; 19)	−2.466	7.624	0.103	58	0.918	0.027

## Data Availability

The data presented in this study are available on request from the corresponding author.

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
