# Peer review of "How Much Money Do You Need to Feel Taller? Impact of Money on Perception of Body Height"

_ijerph, 2021, doi:10.3390/ijerph18094533_

Round 1

Reviewer 1 Report

This is an interesting article studying the relationship between possession of money and owns perception of height. The experiments reported in the paper show that position of money leads to an increased height perception compared to the true height measure. The authors argue that the perception of an increased height is due to an increased sense of attractiveness and social status. The experiments appear to confirm correctly two hypothesis set by the authors:

  • H1: money has an overestimation impact on one’s own height evaluation
  • H2: the height overestimation increases with the stimulus size, namely the amount of money

While there is an interesting discussion in the paper, the links to one’s own perception of increased attractivness or increased social status remain more subjective. Could the authors explain

  • How the experiment was not followed by a self-assessed questionnaire of increased perception of attractiveness or increased perception of social status.
  • Could it have been possible, given the consent of the participants, analyse the social-demographic collected data of participants?
  • Can there be other explanations than attractiveness or social distinction? The sense of wellbeing from financial security may be an important explanation.

I believe authors should be more cautious in their concluding remark in line 326 ‘We were able to verify the hypothesis stating that money activation triggers a social distinctiveness motive’ particularly that they have mentioned in the discussion in line 287 ‘Another explanation that we considered treats money as a social signal that directly influences status perception [22], which in turn is also related to height perception bias[7,8]. Although this path is more direct than the one with money as an attractiveness booster, there is not much previous support for it so we treat it as an alternative explanation.’

I have a worry that the participants were giving consent to an experiment with a given purpose, which turns out to have another intent (study of money influence in one’s own perception rather than mathematical abilities and hand–motor coordination). Can the authors explain and include in the paper how the study was approved by the ethical committee and how they commented specifically in presenting the objectives of the experiment to participants.

Author Response

We are grateful for the insightful review and helpful comments. Below are our responses to the specific points mentioned by the reviewer.

  • How the experiment was not followed by a self-assessed questionnaire of increased perception of attractiveness or increased perception of social status.

Our main goal was to verify the effect of money possession on the estimation of own height, which is a manifestation of attractiveness (Fink et al., 2007) but also of status (Wilson, 1968). In our opinion, showing this effect on the level of perception bias shows that it is not only a matter of attitude, but it could influence real-life cognition.  It might be however a good idea to have a separate measure of the intermediate constructs (attractiveness and status), which could be implemented in future studies.

  • Could it have been possible, given the consent of the participants, analyse the social-demographic collected data of participants?

This is a valid point. Although we did not assume any differences in the socio-demographic data, it could be illustrative to analyse them, if a broad sample would be researched. We did however run the experiment on a college student sample, which is not that varied in terms of socio-demography.

  • Can there be other explanations than attractiveness or social distinction? The sense of wellbeing from financial security may be an important explanation.

We discussed during the preparation of the article some alternative explanation of the received results. There may be some alternative explanations we did not think of, especially because the money activation topic has a broad base of prior research ( as nicely summarized by Wang et al., 2020). We do not think that a sense of well-being is one of those, as the money used in our experiment were of a temporary nature. Specifically, the research participants were never let to believe that they will receive the money, or would be allowed to keep them. But on the other hand, money could trigger a line of thought where participants imagined they would have money in the future. Such an effect would be however rather independent from the amount of money used in the different research groups. On top of that, the discussed amounts (equalling in highest condition to approx. 100 Euro) were not enough for financial security for a month, not to mention longer periods of time. This line of reasoning might be interesting in the context of economical/psychological consequences of unconditional basic income, where economic security would be the main mechanism.  Therefore we think it is a nice idea to follow in future studies, possibly with a different research paradigm.

  • I believe authors should be more cautious in their concluding remark in line 326 ‘We were able to verify the hypothesis stating that money activation triggers a social distinctiveness motive’ particularly that they have mentioned in the discussion in line 287 ‘Another explanation that we considered treats money as a social signal that directly influences status perception [22], which in turn is also related to height perception bias[7,8]. Although this path is more direct than the one with money as an attractiveness booster, there is not much previous support for it so we treat it as an alternative explanation.’

We have rephrased the conclusions more cautiously.

We have demonstrated that in the experimental condition, the participants following the contact with physical banknotes changed their own height perception. It is possibly related to a social distinctiveness motive, which is the main theoretical mechanics assumed for the process. However, we must state that the proposed mechanism is just one of a few possible ones. Some alternative mechanisms include the treatment of money as a social signal directly boosting height estimation or other mechanisms we did not consider. The main research finding concerns the impact of money activation on own height perception.

  • I have a worry that the participants were giving consent to an experiment with a given purpose, which turns out to have another intent (study of money influence in one’s own perception rather than mathematical abilities and hand–motor coordination). Can the authors explain and include in the paper how the study was approved by the ethical committee and how they commented specifically in presenting the objectives of the experiment to participants.

We are highly aware that masking the true purpose of the research at the start of the study was not a fully transparent procedure. We had however not found a viable alternative for the true measurement of the dependent variable (in our case – own height), which would be influenced by our experimental manipulation but not obfuscated by the participants desire to appear honest (which would block their own height boosting propensity). This is why we did include a detailed debrief to the participants right after the experimental procedure, where the true aim of the study was revealed.

We have underlined this in the revised version of the article.

Finally, each person was sent to an adjacent room where they were informed about the real purpose of the study. It was accompanied by an explanation of the reasons for not giving this information at the start of the study. No negative reactions from the participants were observed at this point. In the end, each person was thanked for their participation.

Reviewer 2 Report

While I see some potential interest in this research question, I do not agree that this is of scientific value: Everyone strives for more money, but this is difficult to obtain. Body hight is sth we cannot change, thus, it makes no sense to research this.  Instead of testing

the psychological impact of money on the evaluation of a per- 13
son’s own height

It should better be tested sth like the effect of perceived financial ressources body composition mediating between self-efficacy and contibuting to science, the labor force or combating the covid-19 pandemic...

Furthermore, the manuscript is poorly formatted and doing research with students only is not sufficient. No power calculation and other methodological shortcomings prevent this manuscript to be considered further.

Author Response

We are grateful to the reviewer for honestly stating his or her opinion. We are however not able to agree with the arguments put forward in this review. Below are some of our considerations, which indicate the reasons against the reviewers’ arguments.

 Indeed, people cannot directly alter their height but they may be able to find ways around it. In the case of women, it might be by wearing high heels (Morris et al., 2013; Prokop & Švancárová, 2020), in case of men, it might result in increasing own status, which is in turn related to height evaluation (Wilson, 1968).

 Additionally, we did not test the desire to possess more money, as the reviewer implies, but rather the consequences of activating money, an area that is actively researched by many social scientists (see Wang et al., 2020 for a recent review).

As for the alternative areas of research, suggested by the reviewer, we believe they may be highly interesting for many scientists, but they are not the focus of the current study.

We used a student sample for laboratory research. However, this is not a factor that weakens our results. The use of a wider (more diverse) sample of respondents, using an online questionnaire (currently the most common method) causes many other problems, such as: no base for objective measurement of height, no standardization of height measurement, such as that using a laser, a problem with the effectiveness of on-line experimental manipulation – to name a few. Therefore we believe that our chosen laboratory test method was a good solution, solving some of the most prominent problems appearing in other settings.

Lastly, we need to agree with the observation of the reviewer that we did not conduct an apriori power analysis. A partial reason for it is no previous literature on the postulated effect. However, a post-hoc calculation of the powers shows that the effect is big enough. For experiment 1, a post hoc power analysis with the actual sample size and the eta squared of the main effect in G*Power (version 3.1.9.4), setting α at 0.05, resulting in the estimated effect size (f) of 0.40 and the estimated power (1 − β) which was greater than 0.90. For experiment 2, with the same assumptions, the estimated effect size (f) was 0.62 and the estimated power (1 − β) was greater than 0.95.
Additional argumentation supporting our approach is that we did replicate the effect achieved in sample one in our second study presented in the article.

We hope that the abovementioned arguments may help the reviewer to give our paper a second chance.

Reviewer 3 Report

Thank you for an interesting research paper. I suggest minor revision including the following recommendations:

Please avoid using "we" (first person plural).

The fonts in headings 1.1 - 1.3 are not the same as headings 2.1, 2.2, 3.1, 3.2.

Study 1 and 2: how were the gender distributions in the experimental and control groups?

Table 2: the layout needs improvement, there are not enough space between some of the columns. The control part starts with male data, while the experimental part starts with female data.

Line 201-202: Which group refers to here? “no difference from zero (no overestimation) for either males or females, which finally proves H1”

Line 218-220 and 231-242: unnecessary repetitions of the procedure are observed. An optional suggestion: create a separate section for methodology, where the aim, sampling and procedure for study 1 and 2 can be collected.

The results in table 2 and 3 are not covered enough in the text.

The study didn’t comment on the effect of different banknotes that were included in 200PLN and 400PLN!

Line 315-321 limitations: other factor such as Spatial intelligence can also affect the height estimation, can this count as a limitation of the methodology? Please could you give an explanation regarding this potential issue.

Appendix A and B need to be removed

Are you able to make some general conclusions regarding the effect of contact with money? Many people in the daily life dealing with money, do these peoples feel taller? Can these people be descripted as less accurate in their evaluations?  

Best regards

Author Response

Many thanks for You clear improvement recommendations. We have revised the paper according to Your suggestions. Please find the specific answers below.

  • Please avoid using "we" (first person plural).

We did overview the paper and removed the first-person plural where possible.

  • The fonts in headings 1.1 - 1.3 are not the same as headings 2.1, 2.2, 3.1, 3.2.

The headings fonts were adjusted

  • Study 1 and 2: how were the gender distributions in the experimental and control groups?

We added the gender split in the subgroups by gender in the data presentation (Table 1; Table 3)

  • Table 2: the layout needs improvement, there are not enough space between some of the columns. The control part starts with male data, while the experimental part starts with female data.

Table 2 was adjusted

  • Line 201-202: Which group refers to here? “no difference from zero (no overestimation) for either males or females, which finally proves H1

Line 201-202: we mean here the control group; the text was adjusted

As can be seen from Table 2, there is a significant difference (overestimation) for both males and females from the money activation (experimental) group and no difference from zero (no overestimation) for neither males nor females in the control group, which finally proves H1.

  • Line 218-220 and 231-242: unnecessary repetitions of the procedure are observed. An optional suggestion: create a separate section for methodology, where the aim, sampling and procedure for study 1 and 2 can be collected.

we decreased the procedure description for study 2

  • The results in table 2 and 3 are not covered enough in the text.

As the table two is of a supplementary nature, we did not add an additional description in the text, besides the changes mentioned above. We did modify the text of the paragraph to give better coverage to table 3.

The results shown in Table 3 allow for the second confirmation of H1 – money activation increases one’s own height perception. There is, however, one caveat – the effect did not work for a foreign currency of low nominal value.

The data in Table 3 also proves H2: money with higher nominal value increases one’s own height evaluation more than money of lower nominal value or no money at all. The effect size for a greater amount of money (400 PLN) is bigger than that for a smaller amount (200 PLN).

  • The study didn’t comment on the effect of different banknotes that were included in 200PLN and 400PLN!

We added a comment in the discussion, concerning this effect

It needs to be mentioned that the nominal values of different amounts in various experimental conditions were not achieved with the same banknotes (for example the amount of 200 was composed of two times the 100 note, and 400 - of four times the 100 note). We used a more ecologically true mixture of banknotes, which could have had some impact on the results.

  • Line 315-321 limitations: other factor such as Spatial intelligence can also affect the height estimation, can this count as a limitation of the methodology? Please could you give an explanation regarding this potential issue.

In our opinion, the random procedure of assigning participants to the research groups should equalize the level of people with greater spatial intelligence to those with a lower level of this trait. Methodologically, the distribution of people with different levels of spatial abilities should be the same in experimental and control groups.

As a consequence, there was no room for differences in spatial intelligence to appear in the current study. It is however a variable that may play a decisive role in own height evaluation. Therefore we believe it may be interesting to check in future research.

We added some other potential limitations to the discussion

A further limitation of this study can be the spatial abilities level of our participants, which was not controlled. Future replication should include controlling this trait as well.

  • Appendix A and B need to be removed

 We removed the unnecessary headings

  • Are you able to make some general conclusions regarding the effect of contact with money? Many people in the daily life dealing with money, do these peoples feel taller? Can these people be descripted as less accurate in their evaluations?  

 Such an effect may in fact occur. But it is difficult to observe, as people do not have good means to report or self-present their own estimated height.

We added the following short paragraph to the conclusions section.

An interesting consequence of the study may lay in the real-life application of the results. When money activation biases people’s height estimation, they might make wrong shopping decision (trying on clothes that are too big), misjudge the relative height of others or even act in ways they normally wouldn’t (like flirting with people they would normally not approach due to lower own height evaluation). Although those are only speculations, they could make an interesting basis for further studies.

Round 2

Reviewer 1 Report

This reviewer has observed the updated manuscript and answer from the authors to the review. I have observed that the concluding remarks from the study have been amended according to this reviewer suggestions. I have no further comments.

Author Response

Dear reviewer,
Thank you very much for your initial comments and for accepting the changes we have made to the text.